# Inhibitory Activity of 4-Benzylidene Oxazolones Derivatives of Cinnamic Acid on Human Acetylcholinesterase and Cognitive Improvements in a Mouse Model

**DOI:** 10.3390/molecules28217392

**Published:** 2023-11-02

**Authors:** Alma Marisol Ramírez-Ruiz, Martha Elena Ávila-Cossío, Arturo Estolano-Cobián, José Manuel Cornejo-Bravo, Ana Laura Martinez, Iván Córdova-Guerrero, Bibiana Roselly Cota-Ramírez, Krysta Paola Carranza-Ambriz, Ignacio A. Rivero, Aracely Serrano-Medina

**Affiliations:** 1Facultad de Medicina y Psicología, Universidad Autónoma de Baja California, Calzada Universidad 14418, Parque Industrial Internacional, Tijuana 22424, BC, Mexicoana.laura.martinez.martinez@uabc.edu.mx (A.L.M.); krysta.carranza@uabc.edu.mx (K.P.C.-A.); 2Facultad de Ciencias Químicas e Ingeniería, Universidad Autónoma de Baja California, Calzada Universidad 14418, Parque Industrial Internacional, Tijuana 22424, BC, Mexico; arturo.estolano@uabc.edu.mx (A.E.-C.); jmcornejo@uabc.edu.mx (J.M.C.-B.); icordova@uabc.edu.mx (I.C.-G.); bibiana.cota@uabc.edu.mx (B.R.C.-R.); 3Centro de Graduados e Investigación en Química, Tecnológico Nacional de Mexico/Instituto Tecnológico de Tijuana, Tijuana 22510, BC, Mexico; myf91@hotmail.com

**Keywords:** oxazolones, cinnamic acid, human acetylcholinesterase, inhibitory activity

## Abstract

We synthesized seven (*Z*)-benzylidene-2-(*E*)-styryloxazol-5(4*H*)-ones derivatives of cinnamic acid and evaluated the ability of these compounds to inhibit human acetylcholinesterase (hAChE). The most potent compound was evaluated for cognitive improvement in short-term memory. The seven compounds reversibly inhibited the hAChE between 51 and 75% at 300 μM, showed an affinity (*K*_i_) from 2 to 198 μM, and an IC_50_ from 9 to 246 μM. Molecular docking studies revealed that all binding moieties are involved in the non-covalent interactions with hAChE for all compounds. In addition, in silico pharmacokinetic analysis was carried out to predict the compounds’ blood–brain barrier (BBB) permeability. The most potent inhibitor of hAChE significantly improved cognitive impairment in a modified Y-maze test (5 μmol/kg) and an Object Recognition Test (10 μmol/kg). Our results can help the rational design of hAChE inhibitors to work as potential candidates for treating cognitive disorders.

## 1. Introduction

Many therapeutic drugs for treating different diseases, such as Alzheimer’s disease (AD) [1], insecticides [2], and chemical warfare agents [3], have been synthesized for targeting the inhibition of acetylcholinesterase (AChE). AChE, an α/β-hydrolase fold serine hydrolase, and have evolved to catalyze the termination of cholinergic nerve impulses at a microsecond time-scale, allowing for efficient neurotransmission. As one of the premier biological catalysts, it accelerates the hydrolysis of the neurotransmitter acetylcholine by an estimated 13 orders of magnitude [4]. Human acetylcholinesterase (hAChE) (3.1.1.7) consists of a central 12-stranded mixed β-sheet surrounded by 14 α-helices [5]. The enzyme structure contains three key features: the active catalytic site (CAS), the gorge, and the peripheral anionic site (PAS). The CAS can be divided into regions, including the stearic site, which contains the catalytic triad (Ser203, His447, Glu334), the anionic site (Trp86, Tyr133, Tyr337, Phe338), the oxyanion hole (Gly121, Gly122, Ala204), and the acyl pocket (Phe295 and Phe297). The gorge region is 20 Å deep and 5 Å wide, connects the CAS to the PAS and its compound is primarily aromatic amino acids [6]. Some AChE reversible inhibitors interact at the CAS, such is the case of tacrine (Cognex^®^) which interacts with Trp86, and galantamine (Razadyn^®^, Nivalin^®^) interacts with anionic subsite and with the aromatic gorge. Moreover, donepezil (Aricept^®^) binds the PAS. These drugs have demonstrated therapeutic efficacy on AD patients by enhancing cholinergic transmission. In general, the efficacy of the reversible AChE inhibitors currently available on the market (e.g., donepezil, rivastigmine, and galantamine) is mild and may not be clinically significant due to the gastrointestinal side effects of drugs [7]; tacrine use has been abandoned due to the high incidence of the side effects. Therefore, the search for new drugs implies different therapeutic possibilities.

On the other hand, oxazolones (oxazol-5(4*H*)-ones) or azlactones are five-membered heterocyclic entities containing oxygen and nitrogen in β-position [8]. Functionally substituted oxazolones have been recognized as efficient pharmacophores [9], and their versatility arises from the numerous reactive sites, allowing their chemical modification for diverse applications [10]. Natural and synthetic oxazolones have shown several biological activities such as antimicrobial [9,11], antidiabetic [12], anti-inflammatory [13], tyrosinase inhibitor [14], fungicide, and pesticide [15]. Additionally, several oxazole derivatives have shown functional effects on agricultural pests due to their fungicidal and insecticidal effects with inhibitory doses on the mg/L scale [8]. There are no reports in the literature on the inhibitory activity of oxazolones on hAChE; however, Cavas et.al., synthesized new carbazole-bearing oxazolones and studied the inhibitory activity in vitro on AChE from Electrophorus electricus (EC 3.1.1.7), and the experimental results were modeled using artificial neural network techniques. Oxazolone derivatives inhibited AChE under in vitro conditions, and the authors recommend further investigation with in vitro and vivo studies [5]. Due the importance of these compounds in the present study, we report the synthesis, characterization (see Appendix A), and hAChE inhibition properties of a series of oxazolones derivatives from cinnamic acid. For this series of compounds (Figure 1, (**1**)**–**(**7**)), we have chosen a small dimension benzylidene function to allow its entrance through the cramped catalytic gorge of hAChE. We have added a styryl moiety to optimize the interaction of synthesized compounds with aromatic amino acids characteristic of the PAS and the access gorge. Compounds (**1**)**–**(**4**) were previously synthesized by Mustafa A., et al. [16]. Compounds (**1)**, (**3**), and (**4**) were recently synthetized using K_3_PO_4_ as catalyst [11].

The synthesized compounds could inhibit hAChE through combined interactions: the benzylidene function could interact with the amino acid residues of the catalytic site (anionic subsite); the styryl moiety should also interact with the aromatic amino acid residues of the PAS. The synthesized compounds were evaluated for the hAChE inhibitory activity and studied by UCSF Chimera analysis to find ligand–receptor interactions. Pharmacokinetic and physicochemical properties were determined by in silico predictions. The compound with the best scores was studied to recover drug-induced memory loss in mouse models.

## 2. Results and Discussion

### 2.1. hAChE Inhibitory Screening

The synthesized oxazolones were screened for their inhibition potential against acetylcholinesterase. According to the screening test, which was carried out at an inhibition concentration of 100 and 300 µM, the seven compounds inhibit the enzyme in different percentages. The results are presented in Figure 2.

The percentage of inhibitory activity of the compounds on the human AChE enzyme presented in Figure 2 shows the percentage of inhibition for each compound at a concentration of 100 and 300 µM. The compounds that moderately inhibited the enzyme at a concentration of 100 µM were (**1**) and (**4**) (61.27 and 48.97%, respectively). Increasing the inhibition concentration to 300 µM increases the percentage of inhibition of all the compounds; the order of the inhibition activities of the compounds was (**1**) > (**7**) > (**2**) > (**3**) > (**5**) > (**6**) > (**4**). The results were contrasted with control tests; for this purpose, phosphate buffer solution was used as a negative control and donepezil (DZ), a known anticholinesterase drug, as a positive control. These results show that the functional group present in the benzylidene ring has a vital role in inhibiting acetylcholinesterase. All compounds exhibited lesser activity than compound (**1**) at 100 μM; moreover, at 300 μM they showed moderate activity. Compound (**1**) exhibited the highest activity for both concentrations tested; this may be due to the benzylidene ring’s presence in the oxazole group’s side chain. Compound (**7**) closely resembles compounds (**5**) and (**6**), but the last two lack a -OH group in the benzylidene ring and are weak at 100 μM, whereas it showed moderate activity at 300 μM. Compound (**3**) is inactive at 100 μM, probably due to the chloro substituent in the benzylidene ring; however, at 300 μM, it shows moderate inhibitory activity. Similarly, compound (**4**) is weak at 100 μM due to the methoxy substituent, but at 300 μM it shows less inhibition than compound (**3**). In addition, the methyl group may be another substituent that further activates the oxazolone molecule against acetylcholinesterase activity, as observed in compound (**2**). For acetylcholinesterase inhibitory activity, it may be concluded that a benzylidene group to the side chain of the oxazole group is very much responsible for AChE inhibitory activity. Substituents such as methoxy, chloro, or *ortho*, and *para* phenylacetate may decrease the AChE activity of oxazolone in the present set of compounds.

### 2.2. Evaluation of Kinetics

All the examined oxazolones exhibited reversible inhibition of AChE by forming non-covalent interactions within the enzyme’s active site. We conducted comprehensive enzyme kinetics experiments to determine the binding affinity, represented by the enzyme-inhibitor complex dissociation constant (*K*_i_), of hAChE for compounds (**1**) to (**7**) (as shown in Table 1). The inhibition constants (*K*_i_) for AChE ranged from 2 to 198 µM, with the inhibition potency increasing in the following order: (**1**) < (**7**) < (**3**) < (**2**) < (**4**) < (**5**) < (**6**). AChE demonstrated the highest affinity for compounds lacking substitutions on the benzylidene ring ((**1**)), highlighting the significance of a high lipophilic group for achieving potent inhibition (*K*_i_ = 2.08 ± 0.36 µM). Even when the number of compounds studied can be considered small, we observe certain effects of the substituent groups on kinetics; e.g., the introduction of a hydroxyl group into the phenylacetate ring, as seen in (**7**), resulted in a roughly 33-fold decrease in inhibition potency compared to (**1**) (*K*_i_ = 67 ± 19 µM). Surprisingly, a 1.5-fold increase in AChE inhibition was observed when comparing (**7**) to (**3**), despite a similar size and lipophilicity of the substituents in (**7**) compared to (**5**) and (**6**)**.** It appears that the acetate substituent branching on the benzylidene group in (**5**) and (**6**), as well as the methoxy group in (**4**), reduced inhibition potency towards AChE. The substitution of the methyl group in (**2**) with a stronger electron-withdrawing chlorine group in (**3**) had no significant impact on inhibition, indicating a minor role of these substituents. This is illustrated by the nearly identical *K*_i_ values for corresponding pairs of substituents, i.e., (**3**) vs. (**2**). Our findings align with the observation that AChE displayed the lowest affinity for (**6**), which lacks a lipophilic substituent on the benzylidene group and has approximately 100 times lower inhibition potency than (**1**). The addition of a benzylidene ring to the oxazole group side chain, as seen in (**1**)**,** underscored the importance of having a primary benzylidene group for achieving higher inhibition potency. Studies with a larger number of substituent groups are needed to draw a conclusion. All seven benzylidene-substituted oxazolones reversibly inhibited AChE, displaying different types of inhibition. These results suggest that all seven compounds bind to both the free enzyme (E) and the Michaelis complex (ES). The parameter α > 1 (Table 1) describes the reduction in the ES complex’s affinity for a compound compared to the free enzyme (E) [17]. In most cases, the tested oxazolones bound less strongly to the ES than to the E, except for compound (**6**). When the parameter α equals 1, the inhibitor has equal affinity for the E and the ES complex, indicative of non-competitive inhibition. Alternatively, if α is very small but greater than zero, the inhibitor primarily binds to the ES complex, resembling an uncompetitive model [18]. For most tested compounds, α exceeds one, and the Ks values corresponded to the previously determined Michaelis–Menten constant (Km) [17], implying competitive inhibition and binding of the tested compounds to AChE’s catalytic site. In the case of a compound with an ortho phenyl acetate group ((**5**)), α equals one, indicating no competitive binding with the substrate in the catalytic site and a non-competitive form of inhibition (i.e., binding of these compounds in the PAS). For compound (**6**), we observed an α value very close to zero but greater than zero, suggesting an uncompetitive inhibition mode due to its binding to both the catalytic site and PAS of AChE.

### 2.3. Estimation of IC_50_ Values

To study the potency, the oxazolones with inhibition greater than 50% were selected, taken from the screening test results at 300 μM. The inhibitory potency of seven reversible AChE ligands was investigated by determining the IC_50_, values summarized in Table 1. All ligands in vitro inhibited AChE in a concentration-dependent manner, ranging from IC_50_ values of 9 to 246 μM. From the results, (**1**) (9.2 ± 2.3 μM) was the most potent enzyme inhibitor, while (**6**) (246.3 ± 51.2 μM) was the weakest hAChE-inhibitor of all tested compounds. Comparable IC_50_ values were recorded for the following compounds with the potency in the order: (**1**) > (**7**) > (**3**) > (**2**) > (**4**) > (**5**) > (**6**). 

### 2.4. Molecular Modelling

Docking analysis was conducted to examine the spatial interactions between the synthesized compounds and the enzyme. Initially, each compound derived from oxazolone was docked against the entire three-dimensional structure of AChE, yielding corresponding docking scores. These scores typically represent negative values that signify the binding energy between the compounds and the enzyme. A more negative score indicates a stronger binding interaction, requiring less energy to maintain the molecule–enzyme connection. In our docking algorithm, these scores are associated with binding energy measured in kcal/mol.

Subsequently, a second docking analysis was carried out, focusing on the active site of AChE, as enzyme inhibitors typically act in this specific location. The docking scores for both analyses are presented in Table 2, with each docking validated using the RMSD value (Root mean square deviation of atomic positions) in comparison to the co-crystallized ligand, donepezil, as outlined in the docking methodology. For the docking involving the entire enzyme structure, the RMSD between the docked donepezil and the original one was 1.094. In contrast, for the docking within the selected active site, the RMSD was 0.467. Both of these low RMSD values indicate that the methodology employed is suitable for predicting ligand poses [19].

The docking analysis encompassing the entire enzyme structure revealed the ligand poses within the active site. This explains why, as shown in Table 2, the docking scores for each ligand in both analyses are quite similar, with the largest difference observed for compound (**1**) (a 0.8 kcal/mol difference between the two docking results). Typically, a docking score below −10 kcal/mol is considered indicative of high ligand–enzyme affinity. Notably, all the synthesized ligands met this threshold, signifying their potential as promising binding compounds against hAChE. The most favorable results were obtained for compounds (**2**) with a docking score of −11.5 kcal/mol, (**3**) with −11.4 kcal/mol, and (**4**) and (**1**), both with −11.3 kcal/mol. In comparison, donepezil, a known AChE inhibitor, underwent docking for validation purposes, resulting in docking scores of −12.1 and −12.3 kcal/mol in the analyses involving the complete enzyme and the restricted active site, respectively.

Considering that (**1**) demonstrated one of the best docking results and exhibited the highest activity in vitro among the synthesized compounds, we will delve deeper into its interactions within the AChE active site during the docking process. Figure 3 provides a visual representation of certain amino acid residues within the active site gorge that engage with (**1**). Notably, the surface hydrophobicity of the enzyme gorge reveals that the distal aromatic ring of (**1**) is situated within a relatively hydrophobic region in close proximity to amino acids such as Tyr341, Phe295, Phe297, Phe338, Trp286, and Val294.

Figure 4 highlights a crucial interaction where (**1**) engages in π–π stacking with the aromatic rings of Trp86, maintaining a distance of 3.86 Å. Trp86 plays a pivotal role in interactions with various molecules, including the quaternary groups of acetylcholine and other inhibitory ligands [20,21]. To identify this interaction between the docked pose of (**1**) and AChE, we utilized the hydrogen bond detection tool of UCSF Chimera. While no direct bonds were detected by the software, it is worth noting that the carbonyl group of (**1**) appears to be in close proximity to the hydroxy group of Tyr337, even though the angle and distance (3.93 Å) do not align with the ideal parameters for a hydrogen bond. Notably, the lactone oxygen in (**1**) is at a reasonable distance of 2.94 Å from the hydroxy group in Tyr341, a typical distance for such an interaction. It is possible that the angle between these groups is the reason why no hydrogen bond was detected.

Recent studies on the crystallographic structure of human AChE concur that the enzyme’s active site comprises aromatic residues in two binding sites for ligand binding. The Peripheral Anionic Site (PAS), located at the entrance of the gorge, is considered an allosteric site that guides choline esters towards the catalytic site [22]. It consists of the residues Tyr72, Asp74, Tyr124, Tyr341, and Trp286. The active catalytic site, or esterasic sub-site, is positioned at the gorge’s bottom and includes the catalytic triad responsible for acetylcholine hydrolysis: Ser203, His447, and Glu334. Adjacent to it is the Catalytic Anionic Site (CAS), where the quaternary group of ACh mainly interacts through π–cation interactions, featuring residues Trp86, Tyr133, Tyr337, and Phe338. The oxyanion hole, composed of Gly121, Gly122, and Ala204, interacts with the oxygen in ACh’s carbonyl group, stabilizing the transition states during acylation and deacylation. Additionally, the acyl pocket (Phe295, Phe297, and Trp236) plays a role in stabilizing ACh’s acetyl group [23,24].

Computational studies and X-ray crystallography have proposed the existence of three regions involved in eliminating the hydrolysis products of ACh, contributing to the enzyme’s rapid hydrolysis. These regions include the “backdoor” (comprising Trp86, Gly448, and Tyr449), the “side door” (consisting of Asp74, Asn87, Leu76, and Met 85), and the “acyl loop door” (including Trp236, Arg247, and Phe297) [25,26,27].

The docking results obtained for compound (**1**) in this study suggest that the π–π stacking interaction observed with Trp86 of AChE likely modulates the activity of the enzyme’s “backdoor”, potentially inhibiting its catalytic function. These findings hold promise for the rational design of AChE inhibitors, which could be developed as treatments for Alzheimer’s disease or as insecticides.

### 2.5. In Silico Assessment of Blood–Brain Barrier (BBB) Permeation

Since crossing the blood–brain barrier (BBB) is crucial for identifying promising compounds with central nervous system (CNS) activity, we utilized computational methods to predict the BBB permeability of the tested substances. Table 3 outlines six physicochemical properties of these compounds concerning established benchmarks for CNS-active drugs. Such drugs typically possess low molecular weights (MW < 450), moderate hydrophobicity (logP < 5), a limited number of hydrogen bond donors and acceptors (HBD < 3 and HBA < 7), a restricted number of rotatable bonds (RB < 8), and lower polarity (polar surface area PSA < 70 Å^2^) [28].

In silico forecasts indicate that the tested compounds exhibit favorable attributes, including appropriate lipophilicity (logP = 3.2–4.20), molecular weights within the range of 275.30–349.34 g/mol, polar surface areas up to 47.89 Å^2^, with the exception of compound (**7**) (85.19 Å^2^), and acceptable counts of H-bond donors (max 1) and H-bond acceptors (max 6), along with suitable molecular flexibility (3 to 5 rotatable bonds). The computational predictions suggest that these compounds are likely to traverse the BBB, a desirable feature for their potential AChE inhibition activity and, consequently, their anti-Alzheimer’s profile.

One of the widely adopted criteria for assessing the drug-like properties of a molecule is Lipinski’s rule of five [29], which relates these properties to the aforementioned descriptors, namely lipophilicity, molecular weight, and the number of H-bond donors and acceptors. All compounds exhibiting activity in this study comply with these parameters according to Lipinski’s rule. Additionally, high gastrointestinal absorption is predicted for all compounds.

The in silico assessment indicates that the evaluated compounds are not substrates for P-glycoprotein (P-gp). Since this glycoprotein is associated with expelling xenobiotics from barriers such as the BBB, the synthesized products are less likely to be effluxed, thus aiding their AChE inhibition activity.

Regarding the prediction of cytochrome P450 enzyme inhibition, which is linked to potential drug–drug interactions, the compounds are forecasted to be inhibitors of CYP1A2 and CYP2C9 (except for compound (**7**)), while they are not expected to inhibit CYP2D6 or CYP3A4, the enzymes responsible for metabolizing many well-known drugs.

### 2.6. In Vivo Learning and Memory Tests

#### 2.6.1. The Spontaneous Alternation Test

We used validated mouse models applied by Arunrungvichian et al. [30] to evaluate the agonist activity-substituted 1,2,3-triazoles for nicotinic acetylcholine receptors. Previous to the current work, we applied the testing methods to determine the effective doses of omega-3 fatty acids, EPA, and DHA, for cognition improvement in rats [31].

In our case, oxazolone (**1**) was initially screened to determine its effects on the willingness of rodents to explore new environments at all tested doses using the Y-maze spontaneous alternation test. A slight increase in the activity was observed with an increase in the dose of oxazolone (**1**) (Figure 5), being as active as the control and donepezil. One-way ANOVA with Fisher’s LSD post-hoc comparison was used to evaluate the difference between the dose test and vehicle groups (F_6,28_ = 1.763, *p* = 0.1433, n = 5 mice/group). No statistically significant differences were observed between the test groups; between doses of 5 and 10 μmol/kg, there were no significant differences (*p* = 0.778), and neither was there a significant difference between the dose of 50 and 100 μmol/kg (*p* = 0.925). The minimum effect of oxazolone (**1**) on locomotor activity was significant for 5 μmol/kg (31.8 ± 2.1 entries). In comparison, the maximum effect of the compound was significant for 50 μmol/kg (34 ± 2.3 entries), respectively, compared to the control (38 ± 3.8 entries; *p* > 0.05). No significant differences were observed in the doses of 50 and 100 μmol/kg (*p* = 0.851, *p* = 0.778, respectively) compared with donepezil. According to these results, we ensure that the mice maintain motor coordination and, thus, their ability to navigate the maze and ensure that the treatments did not cause ataxia or sedation.

#### 2.6.2. Spatial Working Memory

Spatial working memory was assessed using the modified Y-maze test [31,32]. In this test, rodents tend to explore new arms of the maze instead of revisiting previously visited ones [33]. The results are depicted in Figure 6. The dosage of donepezil was chosen based on prior rodent studies [34,35]. Statistically significant differences in mean values were observed among the treatment groups when compared to the scopolamine-treated group (F_7,28_ = 0.478, *p* < 0.001, n = 5 mice/group). An increase in the exploration of novel (unfamiliar) arms, compared to the scopolamine group, suggests that both donepezil and oxazolone (**1**) compensated for cholinergic deficits and enhanced spatial working memory. Throughout all sessions, the exploration of unfamiliar arms was significantly higher in the control group compared to the scopolamine group (*p* < 0.001), indicating that the mice were experiencing memory impairment. The cognitive decline induced by scopolamine was notably reversed in the donepezil group, as evidenced by a higher percentage of exploration of novel arms compared to the memory-impaired group (*p* < 0.001). Meanwhile, oxazolone (**1**) significantly improved cognitive deficits at doses of 25, 50, and 100 μmol/kg (*p* = 0.001, respectively). The results indicated that the improvement in cognitive deficits with 25 μmol/kg of oxazolone (**1**) was similar to that achieved with donepezil (*p* = 0.331). However, there were no significant differences observed between oxazolone (**1**) doses of 5 and 10 μmol/kg, 10 and 25 μmol/kg, and 5 and 25 μmol/kg (*p* = 0.818, *p* = 0.349, and *p* = 0.241, respectively). Nevertheless, the disparities in mean values between the 50 and 100 μmol/kg groups were not substantial, and there was no statistically significant distinction either (*p* = 0.066). The minimal effect of oxazolone (**1**) on spatial working memory was significant at 25 μmol/kg (37.83 ± 0.7%), while the maximum effect was observed at 100 μmol/kg (48.30 ± 5.4%), in contrast to the scopolamine group (28.58 ± 5%; *p* < 0.001, respectively). No significant differences were observed at the dosage of 5 μmol/kg compared to donepezil (*p* = 0.055) and 10 μmol/kg compared to donepezil (*p* = 0.085). Consequently, in this experiment, oxazolone (**1**) at doses ranging from 25 to 100 μmol/kg improved spatial working memory in mice.

#### 2.6.3. Recognition Memory

The Novel Object Recognition (NOR) test relies on rodents’ natural inclination to spend more time exploring a novel object compared to a familiar one. This choice to investigate the new object demonstrates the utilization of learning and recognition memory, specifically episodic short-term memory [23]. Initially, a preference test was conducted to confirm that mice did not favor one object over another. The exploration times between the two objects varied significantly for each treatment group (Figure 7), indicating a preference between familiar and new objects. Consequently, the differing exploration times of the two objects (familiar vs. novel) post-treatment reflect the mice’s ability to remember the explored object (Figure 7A).

Mice with memory deficits induced by scopolamine spent an equal amount of time exploring both novel and familiar objects (*p* > 0.05). In contrast, mice in the vehicle group or those treated with donepezil showed significantly different exploration times between objects, indicating a substantial improvement from the memory-impaired state (*p* < 0.01 and *p* < 0.001, respectively). The Discrimination Index (DI) was computed to assess the mice’s capacity to distinguish between familiar and novel objects, with the data presented in Figure 7B. There was a significant difference in DI among the groups (F_7,29_ = 1.74; *p* = 0.010; n = 5 mice/group) compared to the scopolamine group.

A higher DI compared to the scopolamine group indicates compensation for cholinergic deficits and an improvement in episodic short-term memory [30]. The vehicle group exhibited a significantly higher DI (0.478 ± 0.17) than the scopolamine group (DI = 0.180 ± 0.013), demonstrating their ability to differentiate between familiar and novel objects (*p* = 0.002).

Both donepezil at 1 mg/kg and oxazolone (**1**) at doses of 5, 10, 25, and 100 μmol/kg substantially improved episodic short-term memory, as evidenced by increased DI compared to the amnesic group (F_7,29_ = 1.74; *p* = 0.01). The least improvement in episodic short-term memory with oxazolone (**1**) was observed at 10 μmol/kg (DI = 0.317 ± 0.051), while the most significant effect of the test compound was observed at 100 μmol/kg (DI = 0.360 ± 0.056), in contrast to the scopolamine group (DI = 0.180 ± 0.030; *p* = 0.03 and *p* = 0.009, respectively).

However, there were no significant differences between oxazolone (**1**) doses of 10 and 100 μmol/kg (*p* = 0.5), nor were there significant differences between oxazolone (**1**) doses of 10 and 100 μmol/kg and donepezil (*p* = 0.390 and *p* = 0.980, respectively). Therefore, oxazolone (**1**) at doses ranging from 10 to 100 μmol/kg improved episodic short-term memory.

## 3. Materials and Methods

### 3.1. Chemicals

4-((*Z*)-benzylidene)-2-((*E*)-styryl)oxazol-5(4*H*)-one (**1**), 4-((*Z*)-4-methylbenzylidene)-2-((*E*)-styryl)oxazol-5(4*H*)-one (**2**), 4-((*Z*)-4-chlorobenzylidene)-2-((*E*)-styryl)oxazol-5(4*H*)-one (**3**), 4-((*Z*)-4-methoxybenzylidene)-2-((*E*)-styryl)oxazol-5(4*H*)-one (**4**), 2-hydroxy-3-((*Z*)-(5-oxo-2-((*E*)-styryl)oxazol-4(5*H*)-ylidene)methyl)phenyl acetate (**5**), 4-((*Z*)-(5-oxo-2-((*E*)-stiril)oxazol-4(5*H*)-ilideno)metil)fenil acetate (**6**), 3-((*Z*)-(5-oxo-2-((*E*)-stiril)oxazol-4(5*H*)-ilideno)metil)fenil acetate (**7**), were synthesized. Structures of tested oxazolones are given in Figure 1. For more detailed information about the synthesis of (**1)**–(**7**), please refer to the synthesis and characterization section. hAChE was purchased from Sigma Aldrich, Toluca, Mexico. All tested oxazolone solutions were diluted in ethanol and stored at 4 °C before each experiment. 5,5’-dithiobis-(2-nitrobenzoic acid) (DTNB), acetylthiocholine iodide (ATCh), donopezil and bovine serum albumin (BSA), and scopolamine (SC) were also purchased from Sigma Aldrich, Mexico. FTIR spectra were recorded on a Perkin Elmer (Hong Kong) FT-NIR Spectrum 400 equipment. ^1^H and ^13^C NMR spectra were acquired in a Variant Mercury model, operating at 400 MHz for ^1^H and 100 MHz for ^13^C. Tetramethylsilane was used as the internal standard. Mass spectrometry spectra were obtained in a Hewlett Packard (Palo Alto, CA, USA) series II 5890. Uncorrected melting points were determined in a Electrothermal GAC 88629 device (Thermo Fisher Scientific, Waltham, MA, USA).

### 3.2. Synthesis and Characterization of Oxazolones

The synthesis of the oxazolones took place under the classical conditions for the formation of the oxazolones, which is represented in Figure 8.

Cinnamoylglycine (10) [11,36]: Triethylamine (13 mL) was dissolved in anhydrous ether (223 mL) followed by the slow addition of trans-cinnamic acid (10.0 g, 0.067 moles) in a 500 mL in a beaker equipped with a stirrer and cooled with an ice bath to 0 °C. Then, ethyl chloroformate (8 mL) was subsequently added, and the mixture was stirred for 30 min, filtered, and kept the liquid to be added dropwise to a solution of glycine (6.0 g, 79.0 mmol) in dibasic sodium phosphate buffer (50 mM, 80 mL) containing ethanol (40 mL) and ethyl acetate (40 mL). The reaction solution was vigorously stirred for 1 h, and the pH was maintained at 9.0 by adding KOH (50%). The organic solvents in the reaction solution were then removed under a vacuum, and the aqueous phase was acidified to pH 2.0 with hydrochloric acid dropwise. The residue was filtered and washed with ethyl acetate (30 mL × 5). The total yield was 57% of a white product. FTIR (ATR, neat) δ 3282 (NH, amide), 3083 (C-H, aromatic), 2943 (C-H), 1733 (C=O), 1644 (C=O, amide); ^1^H NMR (400 MHz, CDCl_3_) *δ* 8.45 (t, *J* 6.0, 1H), 7.49 (m, 5H, Ar), 6.74 (d, *J* 16.2, 1H, CH), 6.54 (d, *J* 16.2 Hz, 1H, CH), 4.07 (d, *J* 6.0 Hz, 1H, NH), 2.46 (s, 2H, CH); ^13^C-NMR (50 MHz, CDCl_3_) *δ* 176.5, 170.5, 144.5, 140.0, 134.7, 134.1, 132.8, 126.8, 46.0; EM *m*/*z*; C_11_H_11_NO_3_, [M-H]- 204.7.

General method of azlactone synthesis.

*4-((Z)-R-benzylidene)-2-((E)-styryl)oxazol-5(4H)-one* ((**1**)**–**(**7**)) [11,16,36]: Cinnamoylglycine (2.0 g, 9.7 mmol), sodium acetate (0.69 g, 8.5 mmol), the corresponding aromatic aldehyde (11.5 mmol), and acetic anhydride (4.5 mL) were refluxed for 1 h. Following the removal of solvent, the residue was washed with cold ethanol and dried to give a yellow solid.*4-((Z)-benzylidene)-2-((E)-styryl)oxazol-5(4H)-one* (**1**). Yellow solid, yield 34%; mp 124–126 °C; R*_f_* = 0.64 (PE: DCM) (1:1); FTIR (ATR, neat): ν/cm^−1^ 3056 (CH, Ar), 3028 (CH, Ar), 1783 (C=O, lactone), 1648 (C=N, imine); ^1^H-NMR (400 MHz, CDCl_3_) *δ* 8.13 (m, 2H, Ar), 7.71 (d, *J* 16.0 Hz, 1H, CH), 7.59 (m, 2H, Ar), 7.44 (m, 6H, Ar), 7.20 (s, 1H, CH), 6.83 (d, *J* 16.0 Hz, 1H); ^13^C-NMR (100 MHz, CDCl_3_) *δ* 167.3, 163.4, 143.9, 134.6, 133.6, 132.3, 131.3, 131.1, 130.7, 129.1, 128.9, 128.2, 113.4; EM *m*/*z*; C_18_H_14_NO_2_, [M+1]^+^ 276. HRMS (EI) *m*/*z* calcd. For C_18_H_14_NO_2_ [M + 1]^+^ 276.1025, found 276.10245*4-((Z)-4-methylbenzylidene)-2-((E)-styryl)oxazol-5(4H)-one* (**2**). Yellow solid, yield 34%; mp 156–158 °C; R*_f_* = 0.67 (PE: DCM) (1:1); FTIR (ATR, neat) ν/cm^−1^ 3028 (CH, Ar), 1778 (C=O, lactone), 1655 (C=N, imine), 1367 (C-H, CH_3_); ^1^H-NMR (400 MHz, CDCl_3_) *δ* 8.03 (d, *J* 8.0 Hz, 2H, Ar), 7.69 (d, *J* 16.0 Hz, 1H, CH), 7.58 (m, 2H, Ar), 7.43 (m, 3H, Ar), 7.26 (d, *J* 8.0 Hz, 2 H, Ar) 7.18 (s, 1H, CH), 6.82 (d, *J* 16.0 Hz, 1H, CH), 2.41 (s, 3H, CH); ^13^C-NMR (100 MHz, CDCl_3_) *δ* 167.1, 162.9, 143.5, 142.0, 134.7, 132.8, 132.3, 131.6, 130.9, 130.6, 129.7, 129.1, 128.1, 113.5, 21.7; EM *m*/*z*; C_19_H_16_NO_2_, [M + 1]^+^ 290. HRMS (EI) *m*/*z* calcd. For C_19_H_16_NO_2_ [M+1]^+^ 290.1181, found 290.11810*4-((Z)-4-chlorobenzylidene)-2-((E)-styryl)oxazol-5(4H)-one* (**3**). Yellow solid, yield 59%; mp166–168 °C; R*_f_*
**=** 0.57 (PE: DCM) (1:1); FTIR (ATR, neat) ν/cm^−1^ 3032 (C-H, Ar), 1784 (C=O, lactone), 1654 (C=N, imine); ^1^H-NMR (400 MHz, CDCl_3_) *δ* 8.08 (d, *J* 8.0 Hz, 2H, Ar), 7.72 (d, *J* 16.0 Hz, 1H, CH), 7.59 (m, 2H, Ar), 7.43 (m, 5H, Ar), 7.13 (s, 1H, CH), 6.82 (d, *J*16.0 Hz, 1H, CH); ^13^C-NMR (100 MHz, CDCl_3_) *δ* 167.05, 163.7, 144.3, 137.2, 134.5, 133.9, 133.3, 132.1, 130.9, 129.5, 129.2, 129.1, 128.2, 113.2; EM *m*/*z*; C_18_H_13_ ClNO_2_, [M+1]^+^ 310. HRMS (EI) *m*/*z* calcd. For C_18_H_13_ClNO_2_ [M + 1]^+^ 310.635, found 310.06348*4-((Z)-4-methoxybenzylidene)-2-((E)-styryl)oxazol-5(4H)-one* (**4**). Yellow solid, yield 66%; mp168–170 °C; R*_f_* = 0.82 (DCM); FTIR (ATR, neat) ν/cm^−1^ 3071 (C-H, Ar), 3029 (C-H, Ar), 1776 (C=O, lactone), 1651 (C=N, imine); ^1^H-NMR (400 MHz, CDCl_3_) *δ* 8.12 (d, *J* 8.8 Hz, 2H, Ar), 7.67 (d, *J* 16.0 Hz, 1H, CH), 7.58 (m, 2H, Ar), 7.42 (m, 3H, Ar), 7.16 (s, 1H, CH), 6.98 (d, *J* 9.0 Hz, 2H, Ar), 6.81 (d, *J* 16.0 Hz, 1H, CH), 3.88 (s, 3H, CH); ^13^C-NMR (100 MHz, CDCl_3_) *δ* 167.6, 162.4, 162.2, 143.0, 134.7, 134.4, 131.4, 130.5, 129.1, 128.9, 128.0, 126.6, 114.5, 113.5, 55.4; EM *m*/*z*; C_19_H_16_NO_3_, [M+1]^+^ 306. HRMS (EI) *m*/*z* calcd. For C_19_H_16_NO_3_ [M + 1]^+^ 306.1130, found 306.11302*3-((Z)-(5-oxo-2-((E)-styryl)oxazol-4(5H)-ylidene)methyl)phenyl acetate* (**5**). Yellow solid, yield 37%; mp152 °C; R*_f_* = 0.97 (DCM); FTIR (ATR, neat): ν/cm^−1^ 3074 (CH, Ar), 3035 (CH, Ar), 1783 (C=O, lactone), 1648 (C=N, imine); ^1^H-NMR (400 MHz, CDCl_3_) *δ* 7.98 (m, 1H, Ar), 7.18 (m, 1H, Ar), 7.72 (d, *J* 16 Hz, 1H, CH), 7.60 (m, 2H, Ar), 7.46 (s, 1H, Ar), 7.43(m, 3H, Ar), 7.23 (m, 2H, Ar), 7.15 (s, 1H, CH), 6.83 (d, *J* 16 Hz, 1H, CH), 2.34 (s, 3H, CH); ^13^C-NMR (100 MHz, CDCl_3_) *δ* 169.2, 167.0, 163.8, 151.0, 144.3, 135.0, 134.5, 134.3, 130.9, 129.8, 129.7, 129.1, 128.2, 124.7, 124.3, 113.3, 21.1; EM *m*/*z*; C_20_H_16_NO_4_, [M+1]^+^ 334. HRMS (EI) *m*/*z* calcd. For C_20_H_16_NO_4_ [M + 1]^+^ 334.1079, found 334.107930*4-((Z)-(5-oxo-2-((E)-styryl)oxazol-4(5H)-ylidene)methyl)phenyl acetate* (**6**). Yellow solid, yield 35%; mp=170 °C; R*_f_* = 0.94 (DCM); FTIR (ATR, neat): ν/cm^−1^ 3029 (C-H, Ar), 3024 (C-H, Ar), 3000 (C-H, Ar), 1706 (C=O, lactone), 1670 (C=N, imine); ^1^H-NMR (400 MHz, CDCl_3_) *δ* 8.17 (d, *J* 8 Hz, 2H, Ar), 7.71 (d, *J* 16 Hz, 1H, CH), 7.59 (m, 2H, Ar), 7.43 (m, 3H, Ar), 7.21 (m, 2H, Ar), 7.16 (s, 1H, CH), 6.82 (d, *J* 16, 1H, CH), 2.32 (s, 3H, CH); ^13^C-NMR (100 MHz, CDCl_3_) *δ* 168.9, 167.2, 163.5, 152.6, 144.0, 134.6, 133.5, 131.3, 130.5, 130.8, 130.0, 129.1, 128.2, 122.1, 113.3, 21.4; EM *m*/*z*; C_20_H_16_NO_4_, [M + 1]^+^ 334. HRMS (EI) *m*/*z* calcd. For C_20_H_16_NO_4_ [M+1]^+^ 334.1079, found 334.10793*2-hydroxy-3-((Z)-(5-oxo-2-((E)-styryl)oxazol-4(5H)-ylidene)methyl)phenyl acetate* (**7**). Yellow solid, yield 35%; mp 160 °C; R*_f_* = 0.10 (DCM); FTIR (ATR, neat): ν/cm^−1^ 3029 (CH, Ar), 1783 (C=O, lactone), 1655 (C=N, imine); ^1^H-NMR (400 MHz, CDCl_3_) *δ* 8.83 (s, 1H, CH=), 8.22 (s, 1H, CH=), 7.77 (d, *J* 15.5 Hz, 1H, CH=), 7.57 (m, 2H, Ar), 7.42 (m, 4H, Ar), 7.31 (d, 1H, Ar), 7.21 (m, 1H, Ar), 6.60 (d, *J* 15.6 Hz, 1H, CH=), 2.42 (s, 3H, CH); ^13^C-NMR (100 MHz, CDCl_3_) *δ* 168.5, 164.8, 157.8, 143.7, 141.6, 137.6, 134.2, 130.5, 129.0, 128.2, 125.3, 125.0, 124.6, 123.2, 123.1, 121.3, 119.7, 20.6; EM *m*/*z*; C_20_H_16_NO_5_, [M+1]^+^ 350. HRMS (EI) *m*/*z* calcd. For C_20_H_16_NO_5_ [M + 1]^+^ 350.1028, found 350.10285

### 3.3. Enzyme Preparation

Recombinant hAChE expressed in HEK 93 cells, a lyophilized powder (Sigma Aldrich, Mexico), was dissolved in TBSA/sodium azide 0.02% and then was stored in aliquots at −20 °C. Each aliquot was diluted as required in a solution of PBS 0.1M pH 7.4/BSA 0.01% on the day of the experiment. 

### 3.4. hAChE Inhibitory Screening

The spectrophotometric method of Ellman [37] was used to evaluate the inhibition of hAChE. The assay was carried out by double beam UV-Vis 6705 Janway spectrophotometer at room temperature, using optical polystyrene cuvettes of 1.0 cm path length (10 × 10 × 45 mm, 340–800 nm optical transparency). Each test compound was dissolved in ethanol. The final content of ethanol was held constant whenever it did not exceed 0.1%, to limit the influence of ethanol on the degree of enzyme inhibition [38]. Aliquots of 970 μL of PBS 0.1 M pH 7.4/BSA 0.01% containing DTNB (300 μM) and 10 μL of hAChE (0.2 μM) were placed in the polystyrene cuvette. An amount of 10 μL of a stock solution to obtain a final concentration of 100 or 300 μM of the tested compound was added. In total, 10 μL of an aqueous solution of acetylthiocoline 10 mM was added to start the reaction. The absorbance of the sample by the generation of the yellow 5-nitro-2-tio-benzoic anion was recorded at 412 nm at room temperature after 1 min. As a control, an identical enzyme solution without the inhibitor was processed following the same protocol to determine 100% of enzyme activity. Donepezil was used as a reference standard inhibitor. Each experiment was repeated at least in triplicate. The capacity of each compound to inhibit hAChE was expressed as percent inhibition calculated using Equation (1):(1)$$Inhibition(\%)=\frac{Ac-Ai\times 100}{Ac}$$
where *Ac* and *Ai* represent the change in the absorbance in the presence of an inhibitor and without an inhibitor, respectively.

### 3.5. Kinetic Determinations

Reversible inhibition assays from all hAChE were measured using a spectrophotometric Ellman assay [37] at room temperature in 0.1 M PBS pH 7.4, containing 0.01% BSA, containing DTNB 300 µM as a chromogen. The equilibrium dissociation constant (*K*_i_) of the hAChE–oxazolone conjugate was determined using different substrate concentrations (s) ranging from 0.05 to 0.200 mM, and at least four different concentrations of the oxazolones (i) were applied at a given substrate concentration in order to obtain hAChE inhibition between 20% and 80% to establish the affinity for each hAChE–oxazolone conjugate combination. The enzyme activity (ν) was plotted versus the substrate concentration [S] analyzed by non-linear regression analysis (Michaelis–Menten plot); data were transformed into a double reciprocal Lineweaver–Burk plot [39]. Plots 1/ν versus 1/[S] were generated by plotting the slope versus the oxazolone concentration and the ordinate versus the oxazolone concentration of the primary plot. The processing of experimental data to define the oxazolones affinity was performed by a linear regression analysis where the y-intercept determines the enzyme-inhibitor dissociation constants (*K*_i_), while the x-intercept determines the enzyme-substrate dissociation constant (*K*_s_). The equation was used assuming that the substrate, due to low substrate concentrations used in experiments, binds only to the catalytic site, while the inhibitor can bind to both sites, catalytic and peripheral [40]. To determine the mechanism of inhibition, we used the dose-response curves fitted using the Mixed Model Inhibition by the constants V_max_ and K_m_ calculated from Michaelis–Menten kinetics that includes competitive, uncompetitive, and non-competitive inhibition as special cases [17,18,41]. The GraphPad Prims 6.0 software, San Diego, CA, USA was used to calculate the kinetics. All data are shown as means of n = 3.

### 3.6. Estimation of IC_50_ Values

The concentrations of test compounds that inhibited the hydrolysis of substrates by 50% (IC_50_) were determined by monitoring the effect of various concentrations of oxazolones in the assays on the inhibition values. To study the potency, the oxazolones with an inhibition greater than 50% were selected, taken from the screening test results at 300 μM. All the inhibition studies were performed at room temperature in 96-microwell plates using Multiskan FC 5111900 (Lab Systems, CA, USA). The DTNB was used for the measurement of hAChE activity. In total, 174 μL in 0.1 M PBS pH 7.4/0.01% BSA, containing 300 μM DTNB final concentration, 2 μL of test-compound solution, and 2 μL of hAChE solution (0.2 μM) were mixed. The reaction was then initiated by adding 2 μL acetylthiocholine (10 mM), and the plate was read immediately. The formation of yellow 2-nitro-5-tiobenzoico acid due to the reaction of DTNB with thiocholine monitored the enzymatic hydrolysis of acetylthiocholine. Measures were at a wavelength of 412 nm at room temperature after 1 min. The IC_50_ values were then calculated using the GraphPad Prism 6.0 software (San Diego, CA, USA). All data are shown as means of n = 3.

### 3.7. Molecular Modelling

Docking was performed with a hAChE structure (PDB ID: 6O4W) downloaded from Protein Data Bank (www.rcsb.org) (accessed on 25 June 2021). The co-crystallized ligand donepezil was removed, and the enzyme structure was prepared with AutoDock Tools 1.5.6 [42] by water molecules remotion, hydrogens addition, and Gasteiger charges calculation. The docking algorithm employed was AutoDock Vina (Trott and Olson 2010), with a grid box of 30 × 30 × 30 Å with x = 87.9541, y = 84.5448, and z = −3.23301 as the center coordinates. For each analysis, ten poses were generated with an exhaustivity of 8. For the docking protocol validation, docking of the previously removed co-crystallized donepezil was performed, and the RMSD of the resulting pose was compared against the original one. Visualization of the docked poses for their analysis to find ligand–receptor interactions was made with UCSF Chimera [43]. In the case of the analyzed compound, its molecular model was generated through UCSF Chimera employing its SMILES string, and the structure was energy minimized with Chimera’s default conditions with MMTK and Antechamber parameters [44]. Rotatable bonds and atomic charges were defined with AutoDock Tools 1.5.6.

### 3.8. In Silico Assessment of Blood–Brain Barrier (BBB) Permeability

The potential for BBB permeability was assessed through molecular descriptors, including the calculated octanol/water partition coefficient (logP), molecular weight (MW), polar surface area (PSA), number of hydrogen bond donors (HBD), number of hydrogen bond acceptors (HBA), and number of rotatable bonds (RB). These descriptors were determined using in silico methods with the SwissADME 2017 platform [45,46]. The obtained results were then compared to the recommended physicochemical properties for effective drugs targeting the central nervous system [47].

### 3.9. In Vivo Cognitive Tests

#### 3.9.1. Animal Model

The experimental protocols were submitted and approved by the Bioethics Committee of the Faculty of Medicine and Psychology, Autonomous University of Baja California University, Tijuana, México (Record No. 666/2021-2). To perform the in vivo cognitive test experiments, male ICR mice eight weeks of age were used. The animals were housed with access to water and food under a 12 h light and dark cycle, with controlled temperature and humidity.

#### 3.9.2. Amnesia Induced Pharmacologically

To carry out the cognition experiments, we used the methods previously described in detail by Arunrungvichian K., et al., 2015, in which they produced pharmacological amnesia using scopolamine in a mouse model [30]. Donepezil hydrochloride (DZ), was used as a positive control (1mg/kg), whereas scopolamine hydrochloride (SC) was used to produce amnesia in mice (1mg/kg) for 30 min prior to start of the experiments, previously dissolved in saline solution. The compound with the best kinetic profile, oxazolone **1** (4-((*Z*)-benzylidene)-2-((*E*)-styryl)oxazol-5(4*H*)-one) was selected for in vivo cognition experiments in a mouse model at the doses 5, 10, 25, 50 and 100 μmol/kg. All compounds were administered by intraperitoneal injection in a volume of 5 mL/kg. Saline solution was used for the vehicle group (SH).

#### 3.9.3. The Spontaneous Alternation Test

The Y-maze spontaneous alternation test is a behavioral test for measuring the willingness of rodents to explore new environments [33]. The maze is a Y-shaped apparatus with three open arms for the animal to explore freely. Some parts of the brain, such as the hippocampus and the prefrontal cortex, are involved in this task. A Y-maze was used to study the influence of oxazolone **1** on this activity. Each mouse was placed at the end of one arm and was allowed to move freely in the Y-maze for 8 min 1 h after injection of compounds (SH, DZ, and compound **1** at different doses). We count as an entry each time mice accessed each arm at least 10 cm [30]. The number of arm entries and triads are recorded to calculate the percentage of alternation.

#### 3.9.4. Spatial Working Memory

We use the maze used in the spontaneous alternation tests except for the use of a partition to close one of three arms. Here, mice, throughout multiple arm entries, should tend to enter a less recently visited arm or an unfamiliar arm of the maze [32]. The mice were injected with compound **1**, SH, DZ, and scopolamine (1 h and 30 min) before starting experiments. Each mouse was placed at the end of one arm and allowed to move freely for 5 min to get familiar with the two open arms. All arms were opened after 30 min of resting to allow entering in all three arms. An entry was counted if mice accessed an arm at least 10 cm [30]. The percentage of unfamiliar arm exploration was calculated as Equation (2):(2)$$Percentage\;of\;unfamiliar\;arm\;exploration=\frac{number\;of\;unfamiliar\;arm\;entry}{number\;of\;all\;arms\;entry}\times 100$$

#### 3.9.5. Recognition Memory

We employed the Novel Object Recognition Test (NOR) to assess recognition memory, a valuable tool for evaluating cognitive impairment in rodents and testing the impact of new drugs on cognition. This test involves an open-field arena containing two distinct types of objects, each with unique colors and shapes for easy discrimination. The basis of this test lies in rodents’ natural inclination to spend more time exploring a new object compared to a familiar one, leading to the calculation of a Discrimination Index (DI) [32].

The testing procedure unfolded as follows: On the first day, the mice were allowed to freely explore the open-field apparatus for 5 min to become accustomed to the environment. On the second day, oxazolone **1**, DZ, and SH were administered 1 h prior, while SC was given 30 min before the acquisition trial. During this trial, the mice were introduced to the apparatus with two identical objects. After spending 10 min in their cages, one of the objects was replaced with a new one, and the mice were granted 5 min to explore these objects during the retention trial. We meticulously recorded the time the mice spent exploring each object, defined as when their noses approached within 3 cm or when they made contact with the objects. The DI was calculated as (TN – TF)/(TN + TF), with TN representing the exploration time of the new object and TF indicating the exploration time of the familiar object [31].

#### 3.9.6. Statistics

Statistics were analyzed using the SigmaStat 4.00 software. All results are represented as mean ± standard error (SE) for each group. A difference with *p* < 0.05 was considered significant. The different results from the number of entries for the spontaneous alternation test, Spatial working memory, and NOR were analyzed with one-way ANOVA, followed by Fisher’s least significant difference (LSD) post-hoc comparison, except for the exploration time between the acquisition trial and retention trial in NOR, which was analyzed by a paired Student’s *t*-test.

## 4. Conclusions

We synthesized and characterized seven oxazolones derivatives from cinnamic acid. Molecular docking using hAChE as the target presents docking scores above the threshold to be considered a high ligand–enzyme affinity. Most of the compounds inhibit hAChE with IC_50_ values ranging from 9 to 246 μM. Compound (**1**) was the most potent inhibitor with a IC_50_ of 9.2 μM. We selected the compound for two cognition tests using a mouse model. The studied compound (**1**) presents enhancement of the memory, after blocking memory with scopolamine at a potent dose. These findings encourage us to continue the search for oxazolone analogs with more potent hAchE inhibitory activities as potential drugs for the treatment of cognitive impairment diseases such as Alzheimer’s and senile dementia.

## Figures and Tables

**Figure 1 molecules-28-07392-f001:**
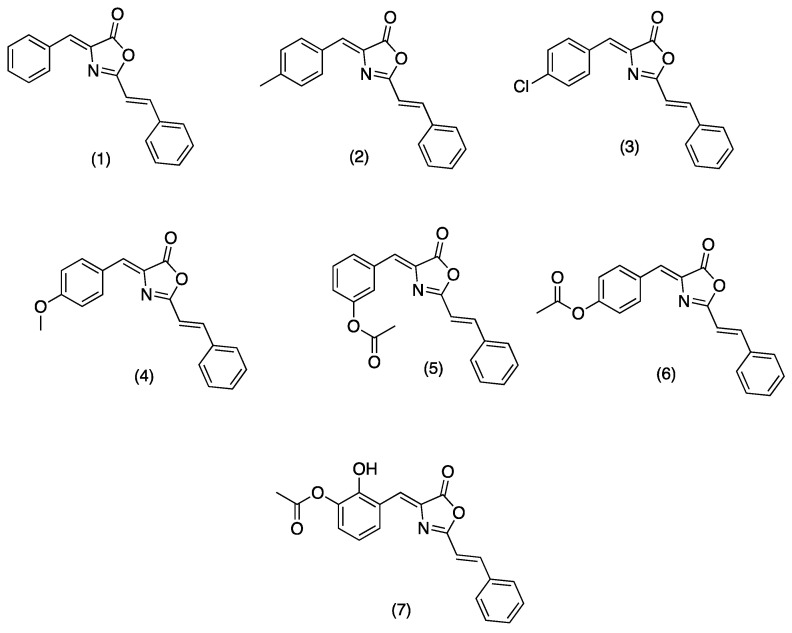
Structure of tested oxazolones or azlactones (oxazol-5(4*H*)-ones).

**Figure 2 molecules-28-07392-f002:**
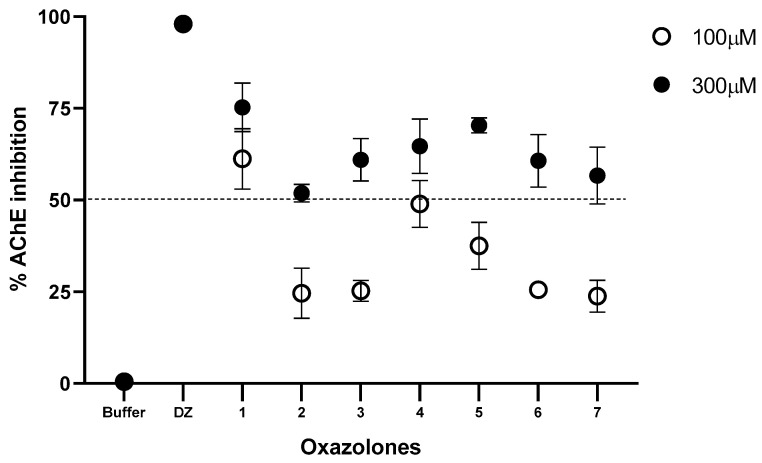
Inhibition (%) of hAChE at 100 and 300 μM for oxazolone tested. Phosphate buffer solution was used as a negative control and donepezil (DZ) as a positive control. All data are shown as means of n = 3.

**Figure 3 molecules-28-07392-f003:**
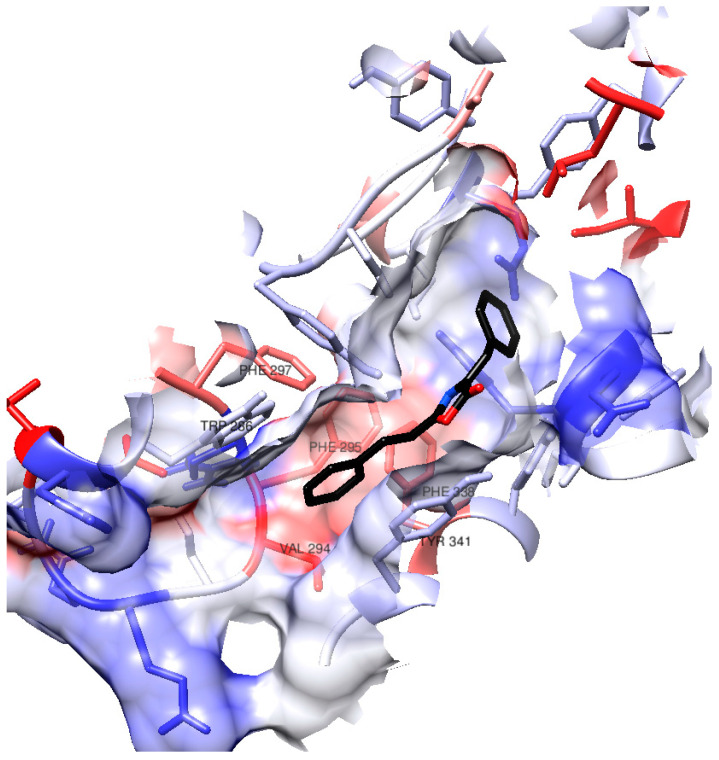
Surface in hAChE active site gorge. The color of the residues’ surface represents their hydrophobicity (blue for hydrophilicity, white as neutral, red as hydrophobicity).

**Figure 4 molecules-28-07392-f004:**
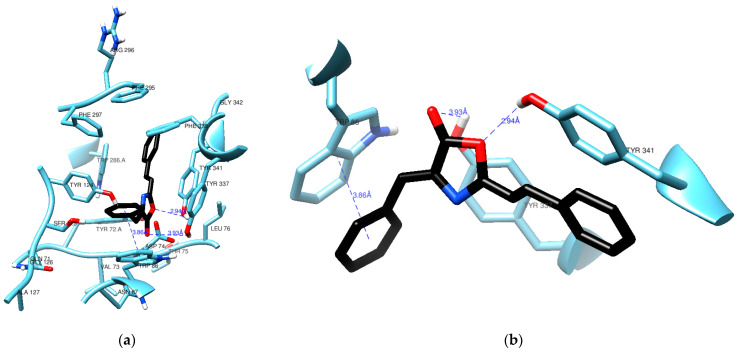
Residues from the active site of AChE in the vicinity (**a**) of compound (**1**) (black) with a clear view of a π-π stacking (**b**) against Trp86.

**Figure 5 molecules-28-07392-f005:**
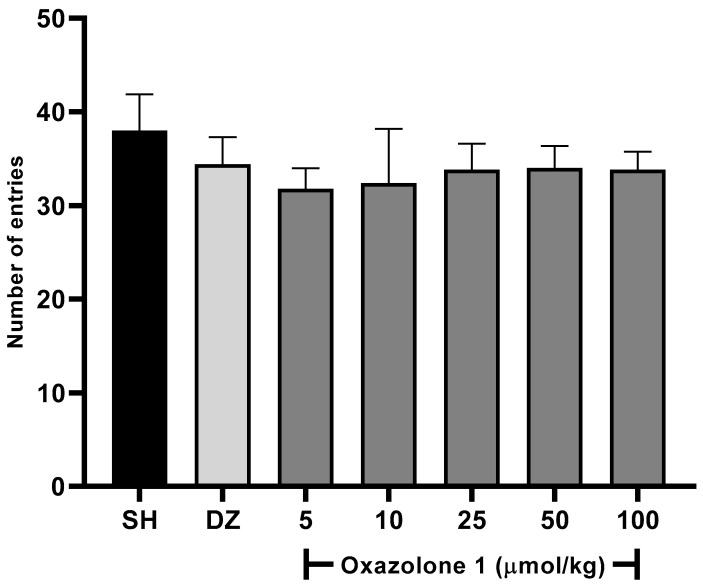
Spontaneous alternation test. There were no significant differences between groups (F_6,28_ = 1.763, *p* = 0.1433), compared to the vehicle group (n = 5 mice/group, one-way ANOVA with Fisher’s LSD post-hoc comparison).

**Figure 6 molecules-28-07392-f006:**
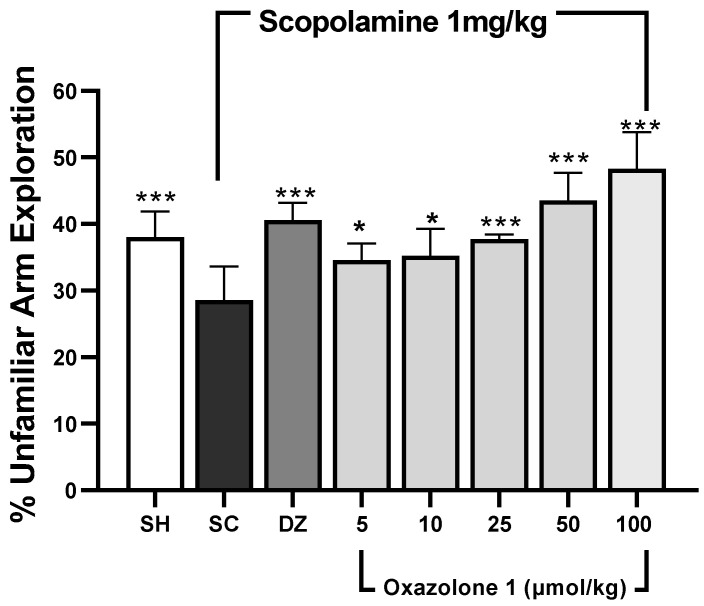
Spatial working memory test (modified Y-maze). Data are represented as mean ± standard error (SE); n = 5 mice/group; * *p* < 0.05, *** *p* < 0.001 vs. scopolamine-treated group (SC), one-way ANOVA with Fisher’s LSD post-hoc comparison.

**Figure 7 molecules-28-07392-f007:**
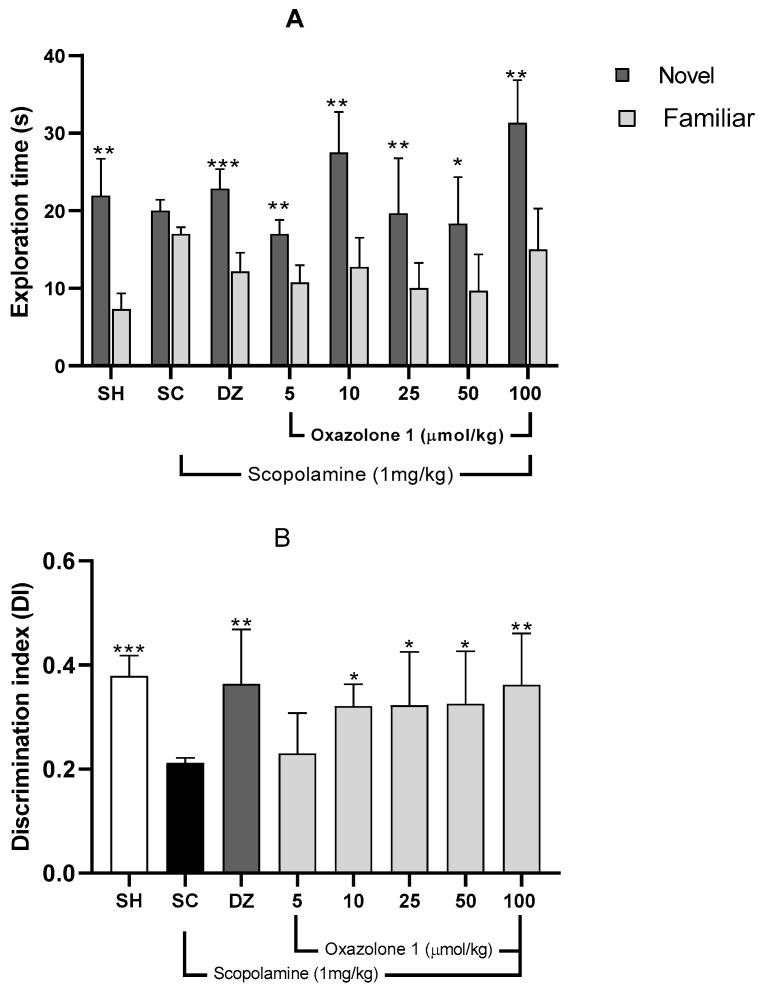
(**A**) Recognition memory evaluation, NOR. Exploration times for familiar (F) and novel (N) objects. Data are presented as mean ± SE; n = 5 mice/group; * *p* < 0.05, ** *p* < 0.01, *** *p* < 0.001 vs. exploration time of a familiar object, paired Student’s *t*-test. (**B**) Discrimination index (TN − TF/TN + TF) in the retention trial of NOR. Data are presented as mean ± SE; n = 5 mice/group; * *p* < 0.05, ** *p* < 0.01, *** *p* < 0.001 vs. scopolamine group (SC), one-way ANOVA with Fisher’s LSD post-hoc comparison.

**Figure 8 molecules-28-07392-f008:**
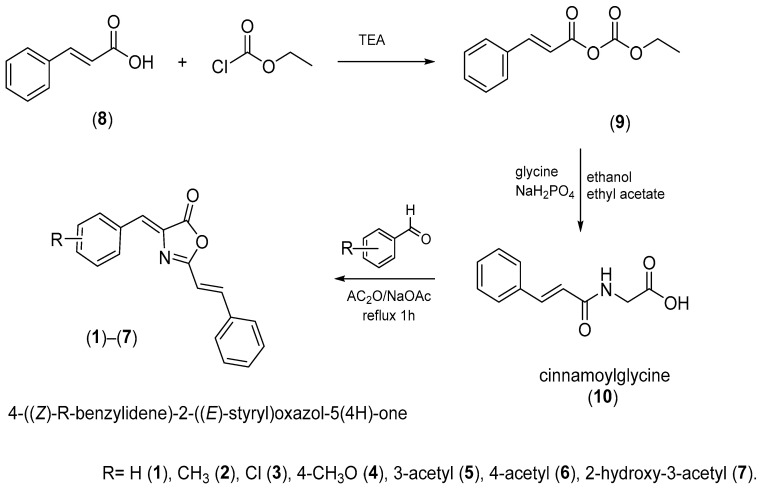
Synthesis of the oxazolones. The compounds were synthesized under mild conditions by condensing cinnamoylglycine and aromatic aldehyde in acetic anhydride as a dehydrating agent and sodium acetate as a catalyst.

**Table 1 molecules-28-07392-t001:** Inhibition constant *K*_i_, IC_50_ values, mode of inhibition and of reversible AChE oxazolones. Parameter α of V_max_ and K_m_ referred to control AChE activity.

Oxazolone	Inhibition Constant (*K*_i_, μM)	IC_50_ (μM)	Parameter α	Mode of Inhibition
1	2.08 ± 0.36	9.2 ± 2.3	21.84	Competitive
2	108 ± 7.77	95.48 ± 1.47	4.98	Competitive
3	98.39 ± 24	89.33 ± 22.33	>1000	Competitive
4	156.9 ± 46	136.8 ± 34.2	4.16	Competitive
5	183 ± 46	203.9 ± 39.7	1	Non-competitive
6	198 ± 49.5	246.3 ± 51.2	0.006	Mixed
7	67 ± 19	37.10 ± 1.61	18.27	Competitive

**Table 2 molecules-28-07392-t002:** Docking score of synthesized compounds with AChE (PDB ID:6O4W).

Binding Energy (kcal/mol)		
Oxazolone	Full Enzyme	Active Site Only
1	−10.5	−11.3
2	−11.5	−11.5
3	−11.4	−11.4
4	−11.3	−11.3
5	−10.9	−11.0
6	−10.8	−11.2
7	−10.3	−10.8
^a^ Donepezil	−12.1	−12.3

^a^ Served as reference compound.

**Table 3 molecules-28-07392-t003:** Pharmacokinetics and drug-likeness of compounds under study calculated by SwissADME.

Descriptors	Oxazolone Compounds						
	(1)	(2)	(3)	(4)	(5)	(6)	(7)
MW	275.30	289.33	309.75	305.33	333.34	333.34	349.34
Num. rotable bonds	3	3	3	4	5	5	5
Num. H-bond acceptors	3	3	3	4	5	5	6
Num. H-bond donors	0	0	0	0	0	0	1
TPSA (Å^2^)	38.66	38.66	38.66	47.89	64.96	64.96	85.19
Consensus Log P	3.67	4.00	4.20	3.66	3.58	3.58	3.20
Log S (Ali)	−4.50	−4.88	−5.16	−4.67	−4.80	−4.80	−4.85
GI absorption	High	High	High	High	High	High	High
BBB permeant	Yes	Yes	Yes	Yes	Yes	Yes	No
P-gp substrate	No	No	No	No	No	No	No
CYP1A2 inhibitor	Yes	Yes	Yes	Yes	Yes	Yes	Yes
CYP2C19 inhibitor	Yes	Yes	Yes	Yes	Yes	Yes	No
CYP2C9 inhibitor	Yes	Yes	Yes	Yes	Yes	Yes	Yes
CYP2D6 inhibitor	No	No	No	No	No	No	No
CYP3A4 inhibitor	No	No	No	No	No	No	No
Lipinski Num. violations	0	0	0	0	0	0	0

MW: molecular weight, TPSA: topological polar surface area, Log P: logarithm of the partition coefficient, Log S (Ali) [28]: model logarithm of molar solubility in water, GI: gastrointestinal, BBB: blood–brain barrier, P-gp: permeability glycoprotein, CYP: cytochrome.

## Data Availability

The data underlying this study are available in the main text and Appendix A.

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
