# Peer review of "Inhibitory Activity of 4-Benzylidene Oxazolones Derivatives of Cinnamic Acid on Human Acetylcholinesterase and Cognitive Improvements in a Mouse Model"

_molecules, 2023, doi:10.3390/molecules28217392_

Round 1

Reviewer 1 Report

Comments and Suggestions for Authors

Although the great work of biological investigations and molecular modeling, the synthesis and characterization are lacking. As a result, I do not propose that this paper be published in molecules in the present form.

Some comments:

1. "We synthesized seven amino-2(5H)-oxazolones derivatives...", lines 15 of page 1. In the article, no such compounds are produced.

2. The letters (E), (Z), and (H) in oxazol-5(4H)-ones should be italicized. Please edit this point throughout the manuscript.

3. There are no references for the synthesis technique of compound 10 and compounds 1-7, as well as previously synthesized compounds such as compound 1 and 10.

4. Where can I find the supporting documentation for the spectroscopic data?

5. The C=O group at 1780 cm-1 for compounds 1-7 is lactone, not ester, as stated by the authors.

6. Please modify the R group in Fig. 8. Some of them are wrong.

7. Lines 434-438 are similar to lines 448-452. Cut out the final one.

8. Where are the other fragments mass in the characterization data section for all produced compounds in the MS?

9. NMR techniques operate at low power levels (200 MHz for 1H and 50 MHz for 13C). 

Reviewer 2 Report

Comments and Suggestions for Authors

Please, see attached pdf-file

Comments on the Quality of English Language

I recommend that the authors show the text of the manuscript to a professional English-speaking chemist to edit its individual parts.

Reviewer 3 Report

Comments and Suggestions for Authors

The manuscript “Inhibitory activity of 4-benzylidene oxazolones derivatives of cinnamic acid on human acetylcholinesterase and cognitive improvements in a mouse model” is devoted to the synthesis of a series of new azlactones and evaluation of their biological activities (in vivo, in vitro, in silico). The performed biological tests are very difficult and important since they include in vivo tests in mice. Thus, the present manuscript is of high practical importance.

In my opinion, this manuscript suits to the scope of Molecules. After major revision, it can be accepted.

However, I have some comments for the authors:

1. Since the authors state that compounds 1-7 are new ones, they must be characterized accordingly, specifically, supporting information must include NMR plots, and their characterization should contain elemental analyses or HRMS.

Comments on the Quality of English Language

Need to check punctuation.

Round 2

Reviewer 1 Report

Comments and Suggestions for Authors

Some comments:

1. In lines 15, page 1, “(Z)-benzylidene)-2-(E)-styryl)oxazol-5(4H)-ones” should be “(Z)-benzylidene-2-(E)-styryloxazol-5(4H)-ones”.

2. In lines 425, page 12, Figure 9should be Figure 1.

3. In page 13, the authors must add a reference for the synthesis of Cinnamoylglycine (10).

4. Regarding compounds 1-7, there are some of them that were previously synthesized. Thus, the authors should add a reference for each compound that was prepared latter. For example, this reference “Mustafa, A. et al. [Justus Liebigs Annalen der Chemie1968, vol. 714, p. 146 - 154]” was prepared many of the seven compounds.

5. Please arrange the characterization data in the supplementary file staring from compound 1, 2, 3, ... etc.

Reviewer 2 Report

Comments and Suggestions for Authors

Dear authors, thank you for your work in revising the manuscript in accordance with the wishes of the reviewers. However, let me draw your attention to the following inaccuracies:

I did not find any additional literary reference to already known previously synthesized compounds (1) – (4), which the authors promised to insert into the text;

The text between lines 440 – 554 again contains duplicate information. Please remove unnecessary parts.

Figure 8. The methylene fragment is not shown in the structural formula of cinnamoylglycine (10)

Ref. 11. Please check the integrity of this reference; the names of the authors and the name of the journal are missing.

In the text of the manuscript authors are encouraged to add a link to the file of supporting information

Reviewer 3 Report

Comments and Suggestions for Authors

In supporting information, NMRs must be integrated and labeled. 
